# Numerical Analysis of Stress Force on Vessel Walls in Atherosclerotic Plaque Removal through Coronary Rotational Atherectomy

**DOI:** 10.3390/mi14122148

**Published:** 2023-11-24

**Authors:** Zhaoju Zhu, Liujing Chen, Weijie Yu, Chuhang Gao, Bingwei He

**Affiliations:** 1College of Mechanical Engineering and Automation, Fuzhou University, Fuzhou 350108, China; chenliujjng5@163.com (L.C.); ywj762485229@163.com (W.Y.); gch141234@163.com (C.G.); mebwhe@fzu.edu.cn (B.H.); 2Research Center of Joint Intelligent Medical Engineering, Fuzhou University, Fuzhou 350108, China

**Keywords:** coronary rotational atherectomy, grinding tool, stress force, numerical simulation

## Abstract

Coronary rotational atherectomy is an effective technique for treating cardiovascular disease by removing calcified tissue using small rotary grinding tools. However, it is difficult to analyze the stress force on vessel walls using experiments directly. Using computational fluid dynamics is a better way to study the stress force characteristics of the burr grinding procedure from a fluid dynamics perspective. For this purpose, physical and simulation models of atherosclerotic plaque removal were constructed in this study. The simulation results show that smaller ratios between the burr and arterial diameter (B/A = 0.5) result in a more stable flow field domain. Additionally, the pressure and stress force generated by the 4.5 mm diameter grinding tool reach 92.77 kPa and 10.36 kPa, surpassing those of the 2.5 mm and 3.5 mm grinding tools. The study has demonstrated the use of computational fluid dynamics to investigate wall shear stress characteristics in medical procedures, providing valuable guidance for optimizing the procedure and minimizing complications.

## 1. Introduction

Atherosclerosis is the primary cause of cardiovascular disease [1]. Calcified tissue is a substance consisting of fat, blood clots, and calcium carbonate deposited in the lining of the arteries [2]. As the atherosclerotic tissue becomes more diseased, it causes the arteries to narrow, restricting blood flow and causing the body to be undersupplied with blood and oxygen. The number of patients with atherosclerotic tissue and complex lesions is growing as the aging population continues to increase [3]. Atherectomy offers unique advantages in treating calcified atherosclerotic tissue. It is an interventional catheter-based procedure that removes atherosclerotic plaque from the arterial wall to restore blood flow and treat cardiovascular disease. During atherectomy, a metal-bonded diamond wheel driven by a long, flexible drive shaft is inserted into the patient’s artery. It rotates at speeds of up to 110,000–170,000 rpm, crushing the plaque into fine fragments that can be absorbed by the blood vessels [4].

Lee et al. [5] marked that rotational atherectomy (RA) and rotational atherectomy (OA) are the primary plaque removal techniques. In cases with large blood vessels or extensive plaque burdens, RA proves to be the optimal choice. Wang et al. [6] showed the need for intensive pretreatment strategies such as upgrading the rotary grinding tool or using a cutting balloon to dilate the vessel after rotary grinding to address severely calcified lesions with a calcification arc of close to 360°. Zhang et al. [7] noted that there are complications but few serious ones with good overall safety and high success rates with coronary artery spinomillography. The American Society of Cardiovascular Angiography and Interventions published a flow chart for the diagnosis and management of calcified lesions in 2020 [8]. In 2021, Chinese interventional cardiologists updated the Chinese Expert Consensus on the diagnosis and management of calcified coronary lesions [9] to better guide the interventional treatment of calcified lesions.

Numerous scholars have conducted studies on the mechanisms of removal of rotational grinding and the improvement of rotor head structure from a simulation perspective, aiming at improving the success rate of rotational grinding. Notably, Computational Fluid Dynamics (CFD) has proven effective in predicting the rotor on-axis flow force [10,11,12]. Feng et al. [13] investigated the internal cylindrical orbital motion with force balance in two dimensions, while Zheng et al. [14] developed a 3D CFD model of rotational atherectomy (RA) to predict the fluid force of a rotating grinding tool operating in an artery, which is difficult to measure experimentally. Kohler et al. [15] conducted computational fluid dynamics analyses and experiments, using graphite as a substitute for calcified tissue, and showed that the magnitude of the rotating grinding force is related to the rotating grinding speed. In clinical settings, the grinding tool is commonly known as a burr for rotational atherectomy. However, since the design scheme of rotary grinding devices mainly derives from clinical data and results, a corresponding theoretical basis is necessary.

Because the design scheme of rotary grinding devices mostly comes from clinical data and results, there needs to be a corresponding theoretical basis. In this study, the changes in the parameters of the flow field and boundary layer during rotational atherosclerotic plaque milling were analyzed from the perspective of hemodynamics through simulation. The numerical simulation method is introduced to analyze the hemodynamic characteristics to predict and guide the surgical treatment plan.

## 2. Materials and Methods

### 2.1. The Geometric Model

During atherosclerotic plaque removal through atherectomy, the coupled motion of the grinding wheel and blood flow in the vessel is a critical factor to consider. The hydrodynamic pressure film between the tool and the plaque is relatively thin, allowing direct contact and grinding of the plaque. Conversely, when the tool encounters the normal blood vessel wall, the dynamic pressure film between the tool and the vessel wall becomes thicker. This thicker film acts as a barrier, preventing the tool from grinding the normal blood vessel tissue. To simplify the rotational flow of the grinding wheel within the artery, an ellipsoidal shape is utilized, and this penetrates deep into the vessel through a guide rod. The resulting flow can be simplified as an eccentric annular flow with internal cylindrical rotation, as shown in Figure 1. These experiments are conducted using arterial columns with diameters of 5 or 6 mm to study coronary or peripheral plaque removal procedures, respectively [16,17]. The catheter used in these experiments is set parallel to the arterial axis, with an ellipsoidal protrusion on the exit end to simulate intravascular plaque deposition. In these experiments, a 0.1 mm clearance is left between the grinding tool and the cylindrical arterial wall to facilitate the removal of plaque from the inner wall of the vessel [18]. The blood flow velocity within the artery is highly rapid, averaging around 18–23 cm/s. The entrance speed is set at 20 cm/s to maintain a stable flow. The initial pressure at the outlet is set to zero to simulate a sufficiently long blood vessel. To study its effects comprehensively, the rotational speed is set to 110,000 rpm, 125,000 rpm 140,000 rpm, 155,000 rpm, and 170,000 rpm. These speeds are chosen based on their relevance to clinical settings and to capture the range of rotational speeds typically used in atherectomy procedures.

In this study, three diameter sizes (2.5 mm, 3.5 mm, 4.5 mm) of rotary grinding tools were selected as rotary grinding tools [19], and their head size parameters and structures are shown in Table 1 and Figure 2. According to the different sizes of vessel diameters (5 mm, 6 mm), four groups were set up for comparison experiments wherein the B/A value is the point we focus on, which is an important factor affecting the stability of the flow field under the working condition of the rotary grinding device [14].

### 2.2. The Numerical Model

The blood flow characteristics during atherosclerotic plaque removal were modeled in ANSYS—CFD. The numerical simulation of this study is based on Navier stokes equation and continuity equation.
(1)$$\rho \left[\frac{𝜕u}{𝜕t}+\left(u\cdot \nabla \right)u\right]+\nabla p-\mu \nabla^{2}u=0\;\nabla \cdot u=0$$

Here, u is the fluid velocity vector, p is the pressure, ρ is the density, and μ is the blood viscosity.

Blood served as the simulation fluid in the research study, with density of 1060 kg/m^3^ and viscosity of 0.003 kg/(m·s). The Taylor number is calculated in this flow dominated by rotation in order to determine the turbulence effect.
(2)$$Ta=w^{2}r{\left(R-r\right)}^{3}v^{-2}$$

Here, w is the burr rotational speed and *r* and *R* are the radius of the burr’s radius at its equator and the arterial wall, respectively. With *r* = 1.25 mm, *R* = 2.5 mm, and ω = 110,000 rpm, Ta = 3.2 × 10^9^, which is much higher than the critical value of turbulence, 1700 [20].

Therefore, the RNG k-e turbulence model is used to solve the cyclonic flow [21]. An enhanced wall function improves the calculation accuracy within the boundary layer. Second-order format discrete pressure, momentum, turbulent kinetic energy, and turbulent dissipation rate are applied. We use the semi-implicit pressure-dependent simple-C equation to handle velocity–pressure coupling. For the residuals of the continuity, momentum, and turbulent transport equations, the convergence criterion is set at 10^−6^. Additionally, convergence is acknowledged if the flow difference between the entrance and exit is less than 1% of the entrance value.

### 2.3. Experimental Setup

In addition to conducting a computational fluid dynamics (CFD) study of plaque spinning, the macroscopic flow inside the aneurysm was also examined using the laser particle image velocity (PIV) field testing technique. The PIV technique is a well-established and widely used full-field, interference-free, transient measurement method for flow velocity in fluid mechanics research. The basic principle of PIV is illustrated in Figure 3. In a PIV experiment, a large number of tracer particles that strongly follow the flow are introduced into the measured flow field. A laser emits a sheet light source onto the region of interest, and after several exposures, motion images of the tracer particles are obtained using a high-speed camera. The instantaneous spatial position changes of the particles are recorded and analyzed to provide velocity information of the flow field. Through this method, we were able to obtain accurate and comprehensive flow velocity data for the aneurysm. These results were then compared with those from the CFD simulations, confirming the reliability of our computational approach.

The grinding experimental platform is shown in Figure 4, and it consists of grinding device, fluid delivery systems, measurement systems, and light source system. During the experiment, the measurement area was illuminated with a cold light source, and the image data were collected using a high-speed camera. To avoid the flocculation of measured tracer particles in the liquid, deionized water was used as a blood substitute in this study and a magnetic stirrer was used to continuously stir. Moreover, the liquid flow rate was set to 30 mL/min. The deionized water with debris was made to flow through a PVC pipe and collected in a reservoir for further analysis.

The images acquired using the camera were transferred to a computer memory system, where the velocity vectors could be calculated and stored directly by the PIVlab software in MATLAB R2021a. The data could also be processed by the third-party post-processing software Tecplot 2021 R1 in order to obtain more aesthetically pleasing data results. 

## 3. Results

Atherectomy is a medical procedure that involves the removal of atherosclerotic plaque from the arterial wall using a rotary grinding tool. This procedure, though effective in removing plaque, has the potential to disrupt flow throughout the flow field domain and cause pressure and stress force on the vessel wall, making it imperative to monitor parameter indicators during the process. However, measuring these indicators directly is difficult. Hemodynamic analysis plays a critical role in understanding fluid trajectories, velocities, and simulated wall pressure and stress changes. Trajectories provide a visual representation of the vector field, enabling a better grasp of the fluid’s motion. Velocity studies are also crucial, as the grinding tool must achieve a high speed to control ground debris and safely pass it through capillaries [22]. The velocity distribution of the flow field can also influence the movement of abrasive debris, especially in low-speed vortex regions where deposition is probable. The abrasive head of the tool operates on one side of the vessel’s inner wall, making it simple to apply pressure to the vessel wall and plaque tissue. Although appropriate pressure and stress force are beneficial in removing plaque tissue, they can also harm the arterial wall. 

### 3.1. Fluid Dynamics Characterization

A cross-section YZ was added inside the simulated vessel, and the cross-section YZ passed vertically through the axes of the vessel and the rotating grinding tool, as shown in Figure 5. By observing the flow velocity distribution on the cross-section YZ, the flow field domain in the motion state of the rotating grinding tool could be analyzed.

The size and rotational speed of the rotating grinding tool affect the flow velocity distribution in the arterial vasculature. Figure 6 shows the velocity clouds for different groups at different rotational speeds, as well as the velocity vector plots.

In Group A of Figure 6, the region with a flow rate surpassing 2 m/s is primarily concentrated near the rotating tool. This observation supports the argument that the rotational motion of the tool acts as the dominant force propelling the flow across the entire flow field. By setting the inlet as a velocity outlet, the simulate fluid diffuses towards the outlet. As there is a gap between the tool and the inner wall of the blood vessel in the radial distant direction, the high-speed region gradually diffuses towards the radial distant end before eventually shifting towards the axial end. The narrow gap between the inner wall on the side of the tool and the plaque prompts the fluid in this region to be swiftly squeezed and diffused, forming a sudden surge in fluid flow rate. The kinetic energy of this fluid is lost upon colliding with the inner wall of the blood vessel, leading to minimal axial velocity diffusion on this side. The existence of the plaque region in the flow field results in the creation of a low-speed region at the gap between the plaque and the grinding head due to the obstructive effect of the plaque. The existence of the plaque region in the flow field results in the creation of a low-speed region at the gap between the plaque and the grinding head due to the obstructive effect of the plaque. The small grinding debris generated at this location during the grinding process forms secondary sedimentation. After the fluid flows through the plaque region, it breaks through the obstruction, causing a slight increase in velocity before the high-speed region gathers again at the bottom of the plaque. The rotational motion of the tool is the crux of the flow field and the presence of the plaque in the flow field has a complex influence on the fluid flow.

A velocity probe line was added to the velocity cloud of cross-section YZ, with the radial direction of the probe line passing vertically through the center point of the grinding tool and the axial direction in the direction of the central axis of the uppermost end of the grinding tool and the vessel wall surface. The velocity value magnitude was exported using Tecplot and processed using Origin to obtain a graph of the velocity magnitude on the probe line. As shown in the analysis presented in Figure 7, the magnitude of the velocity on the detection line is related to the diameter of the grinding tool and container. As the rotation speed increases, the velocity on the detection line also increases, and there is no significant change in the size of the velocity in the axial distance. This indicates that the increase in rotation speed does not extend the high-speed region to the axial direction. The smaller the grinding tool whose diameter is 2.5 mm is, the smaller the peak velocity on the detection line is. For the same diameter of blood vessels and rotational speed, the maximum flow rate that can be reached using a 4.5 mm grinding tool is about twice that of the rate that can be reached using a 3.5 mm grinding tool. This proves that the use of 4.5 mm grinding tools has a strong destructive effect on the flow field. As the diameter of the grinding tool increases, the peak velocity increases correspondingly. This is because the smaller the gap between the grinding tool and the distal vascular wall is, the faster the fluid velocity value increases due to the operation of the rotating grinding tool. When the value of B/A decreases to 0.5, the flow field expands to half of the entire vessel diameter. The radial distance between the grinding tool and the distal blood vessels also increases proportionally, as the grinding tool is placed on one side near the vessel. The ratio of the grinding tool diameter to the container diameter is an important factor affecting the stability of the flow field domain.

Vortex flow is an important factor that must be examined in the flow field. This is because pressure differences can generate vortices when fluid flows along a profile perpendicular to the velocity [23]. The presence of a vortex field can significantly affect the development of the flow field, and the velocity clouds are vectorized to add flow traces. Our observations indicate the presence of four distinct vortices, which included a pair of counter-rotating vortices near the vessel wall adjacent to the grinding tool, two relatively larger vortices located at the upstream and downstream edges of the field, and a vortex region downstream of the plaque due to the obstructive effect of the plaque. The counter-rotating vortex pair have formed in the same manner as the Taylor–Couette flow [24]. The remaining vortices have originated from the four primary vortices. It is noteworthy that the vortex region located downstream at position 5 in the Figure 8 represents shedding, a periodic shedding of vortices caused by fluid on both sides behind the obstructing object. In summary, the examination of vortex flow is crucial for assessing the flow field, particularly when analyzing pressure differences generated by fluid flow along a profile perpendicular to the velocity. The existence of vortex fields has a significant impact on the development of the flow field, as observed through vectorized velocity clouds and flow traces.

As shown in Figure 8, with the same artery diameter, the higher rotational speed increases the vortex size near the rotary grinding tool. It spreads the high-speed region further away from the head. As the flow field changes due to the rotational speed of the rotary grinding tool, the direction and size of the vortex flow located at the plaque change accordingly.

The high-velocity motion of fluids can exert extreme pressure and stress force on the vessel wall, particularly near one side. This can lead to the breakage of the arterial wall, which is strongly associated with high pressure and wall shear stress (WSS). To better understand the dynamics of grinding tool motion and blood flow, it is necessary to explore the pressure and stress force. To do so, the stress force should be selected in the circumferential direction of the vessel wall, with a length equivalent to the circumferential region of the probing line L. Meanwhile, the pressure field should be selected in the cross-section XZ, over the central plane where the axis of the grinding tool is located. Figure 9 illustrates this selection process.

The flow line in vessel grinding, as depicted in Figure 10, assumes a circular shape around the arterial axis due to the exceedingly high rotational motion of the grinding tool.

The flow engenders a high-pressure region prior to entering the gap between the grinding tool and the vessel wall close to one side and a low-pressure region upon its exit from the gap. This pressure difference acts as feedback to drive the orbital motion of the grinding tool, with the minimum local pressure occurring at the narrowest point of the gap. Comparing the pressure field analysis plots across various groups, it becomes apparent that the radial diameter of the grinding tool emerges as the primary factor influencing the pressure field distribution. As Figure 8 portrays, the pressure field of Group B, which employs a 4.5 mm grinding tool diameter, corresponds to a maximum pressure of 92.77 kPa. Conversely, Group C, which operates with a 2.5 mm grinding tool diameter, reports a maximum pressure scale of 23.52 kPa. Groups A and D leverage a 3.5 mm grinding tool in 5 mm and 6 mm vessels, respectively, with their pressure fields indicating fluctuation within the control range. It bears noting that the larger the radial diameter of the grinding tool is, the greater the magnitude of fluid fluctuation is, and consequently, the higher the pressure value exerted on the simulated vessel wall surface is. This adverse effect manifests in the direct rupture of the vessel wall surface. Hence, it proves imperative to consider, carefully, the size of the grinding tool when performing such procedures. This study underscored the pivotal role of the radial diameter of the grinding tool in modulating the pressure field distribution during vessel grinding. Such insights afford valuable guidance toward optimizing vessel grinding procedures while emphasizing the need for appropriate equipment to minimize potential harm.

As shown in Figure 11, the stress field distribution is mainly related to the direction of rotation and the morphology of the grinding tool. Wall shear stress is a crucial factor in arterial rupture, with favorable blood dynamics leading to accelerated plaque tissue damage, while unfavorable biomechanical or physiological factors can ultimately result in arterial wall expansion and rupture [25]. The stress field is mainly concentrated on the side of the closed vessels, and the direction of rotation of the grinding tool affects the development trend of the stress field. As the rotational speed increases, the stress force increases accordingly, and the diffusion area of the stress field increases. The development shape of the stress field is related to the size of the long axis of the grinding tool. With a diameter of 4.5 mm, the grinding tool exhibits a larger volume in comparison to the 2.5 mm and 3.5 mm counterparts. Consequently, the axial diffusion area of the stress field also expands correspondingly.

### 3.2. Experimental Verification

The internal flow field of the coronary rotational atherectomy was studied through PIV testing of the steady-state field according to the experimental system shown in Figure 4. The presented experiment included two figures for data visualization. Figure 12a depicts the images of particles captured by the camera during the experiment. On the other hand, Figure 12b shows the fluid trace image after PIVlab and Tecplot processing. These figures provide valuable insights into the experimental results and highlight the effectiveness of the utilized image processing techniques. The images in Figure 11a allow for a visual analysis of the particle behavior in the fluid flow, while the processed image in Figure 12b provides a more comprehensive understanding of the flow structure and dynamics. The images presented in this experiment serve as evidence of the observed phenomena and provide a foundation for the discussion and analysis of the results. Furthermore, the use of PIVlab and Tecplot processing techniques adds credibility to the experimental data by ensuring accurate and reliable measurements.

The experimental results show that there are four eddies during the rotation of the rotating tool; a pair of reverse rotating eddies exist on the container wall near the grinding tool, and two relatively large eddies with opposite rotation directions exist at the upstream and downstream edges of the flow field. The accuracy of the simulation results has been confirmed.

## 4. Conclusions

This article investigates the use of rotational grinding tools in interventional surgery and applies computational fluid dynamics (CFD) simulations to analyze the fluid dynamics of these tools. The study examined the velocity, vorticity, pressure, and stress force of the boundary layer caused by the rotation of the grinding tool in a simulated container. The purpose was to provide theoretical support for existing studies on rotational grinding for plaque removal and to enhance our understanding of blood flow characteristics during this process. The simulation analysis yielded the following conclusions:(1)As the value of B/A increases, the velocity within the flow field also increases. Due to the constraint of the vessel wall, the direction of velocity diffusion changes from radial to axial.(2)The rotation of the grinding tool generates a vortex region, which increases the possibility of the secondary deposition of ground material during the rotational grinding process.(3)The 4.5 mm diameter grinding tool diameter produces the pressure of 92.77 kPa during the rotational grinding process, which causes the container wall to form roughness.(4)The shape and rotation direction of the grinding tool can influence the distribution and development of the stress force field. The simulation results reveal that the stress force generated by fluid movement is 10.36 kPa, a value considerably lower than the actual measurement.

Overall, this study provided valuable insights into the fluid dynamics of rotational grinding and its effects on vessel walls that can serve as references for the development of safer and more effective surgical tools and techniques.

## Figures and Tables

**Figure 1 micromachines-14-02148-f001:**
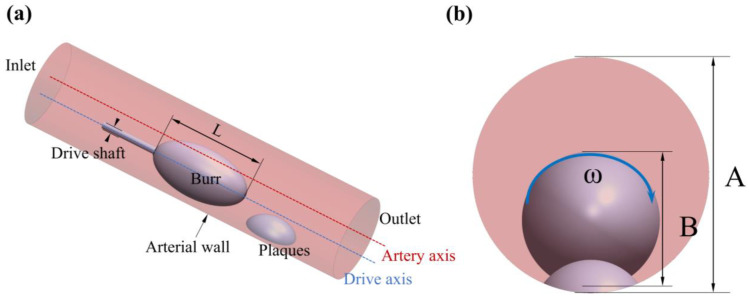
Geometric model of the rotary grinding device. (**a**) axonometric view; (**b**) front view.

**Figure 2 micromachines-14-02148-f002:**
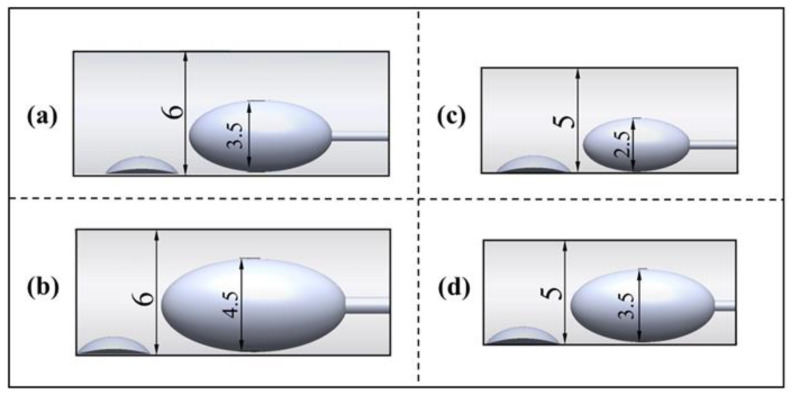
Control chart of experimental groups. (**a**) burr 3.5 mm, vessel 6 mm; (**b**) burr 4.5 mm, vessel 6 mm; (**c**) burr 2.5 mm, vessel 5 mm; (**d**) burr 3.5 mm, vessel 5 mm.

**Figure 3 micromachines-14-02148-f003:**
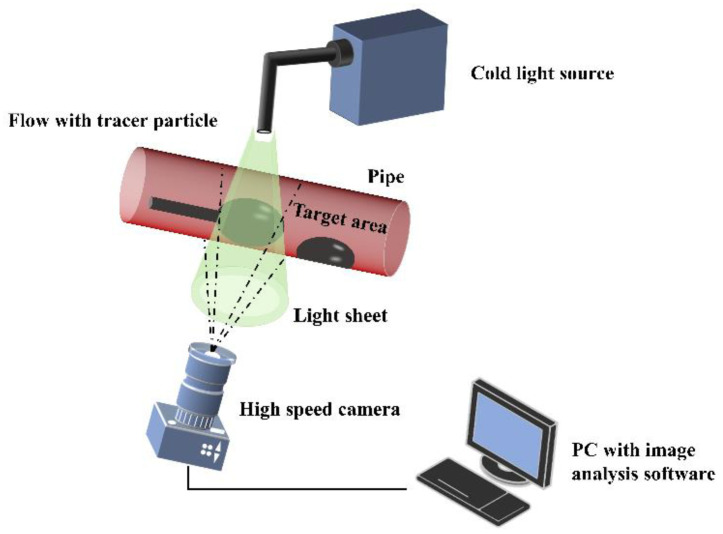
Schematic diagram of PIV experiment.

**Figure 4 micromachines-14-02148-f004:**
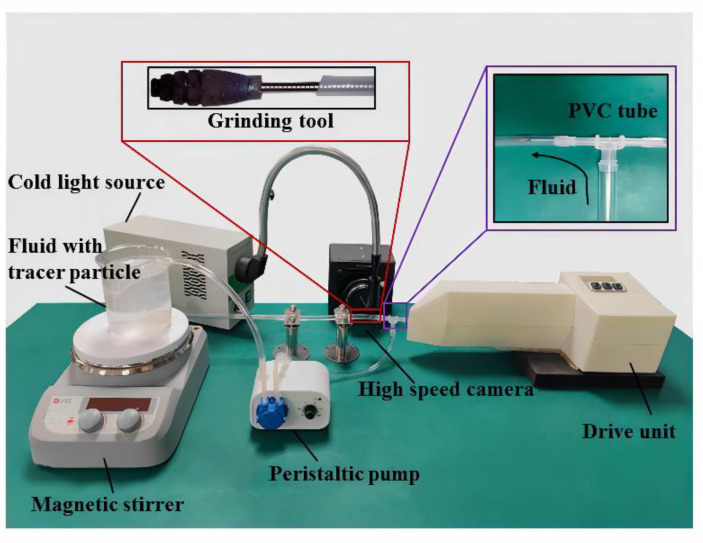
Experimental platform setup.

**Figure 5 micromachines-14-02148-f005:**
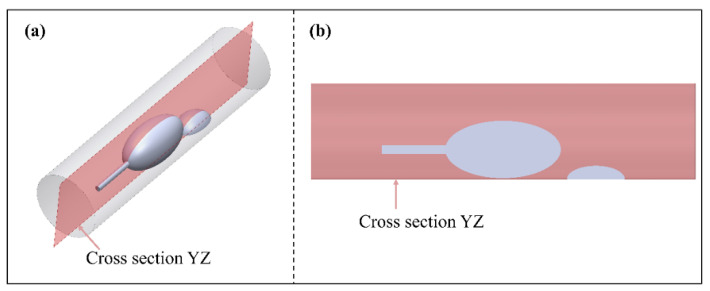
Cross-section of velocity diagram. (**a**) axonometric view; (**b**) front view.

**Figure 6 micromachines-14-02148-f006:**
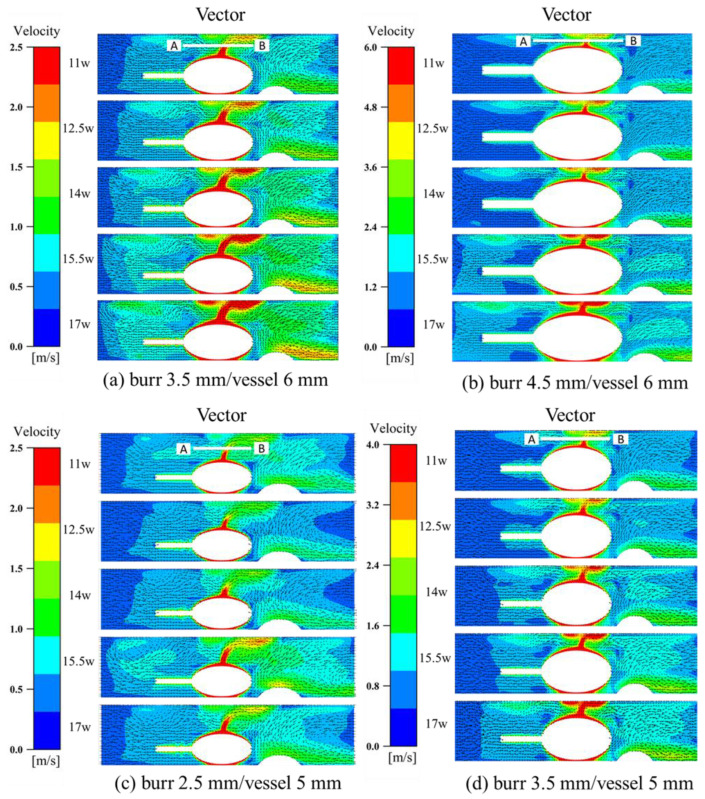
Velocity clouds and velocity vectors in different groups.

**Figure 7 micromachines-14-02148-f007:**
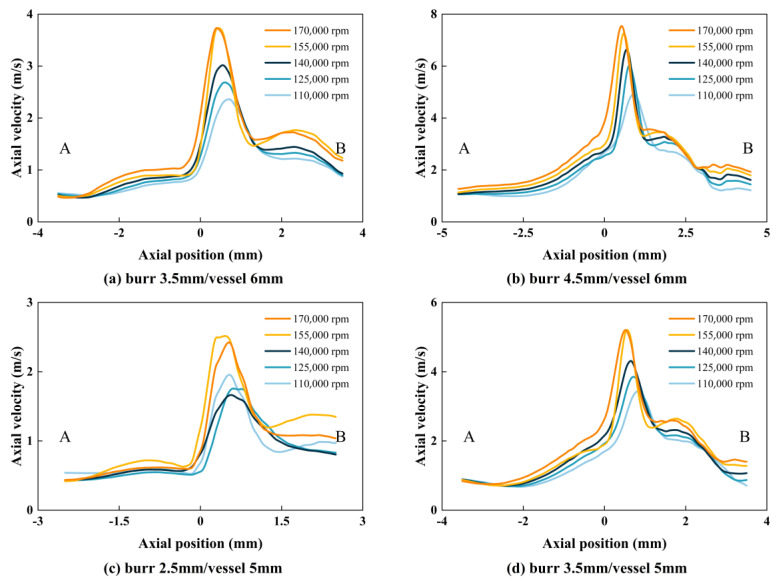
Detection line speed distribution.

**Figure 8 micromachines-14-02148-f008:**
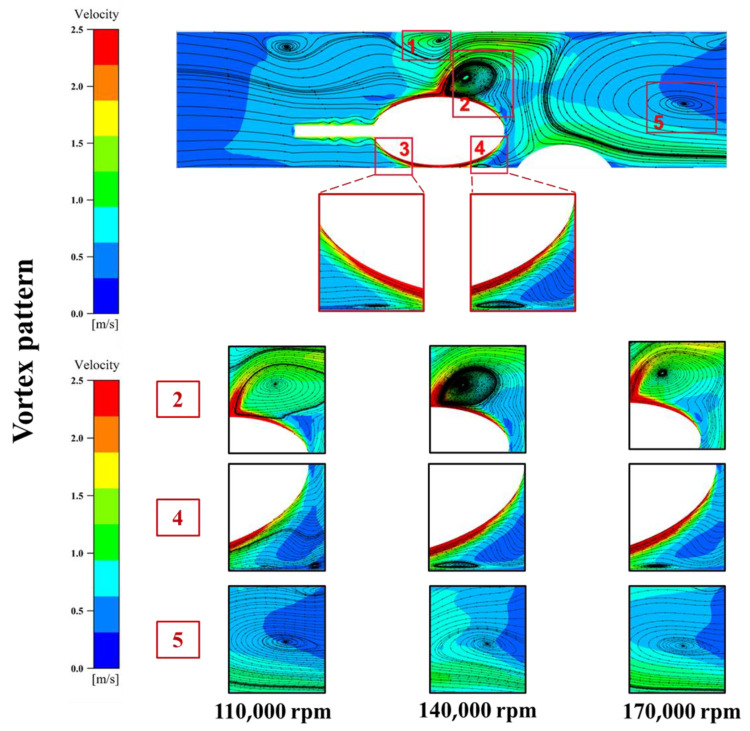
Vortex diagram of the same group with different speed (Positions 1–5 are vortex areas).

**Figure 9 micromachines-14-02148-f009:**
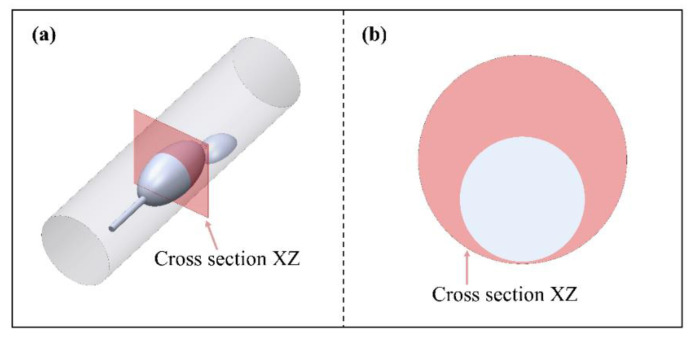
Cross-section of press diagram. (**a**) axonometric view; (**b**) front view.

**Figure 10 micromachines-14-02148-f010:**
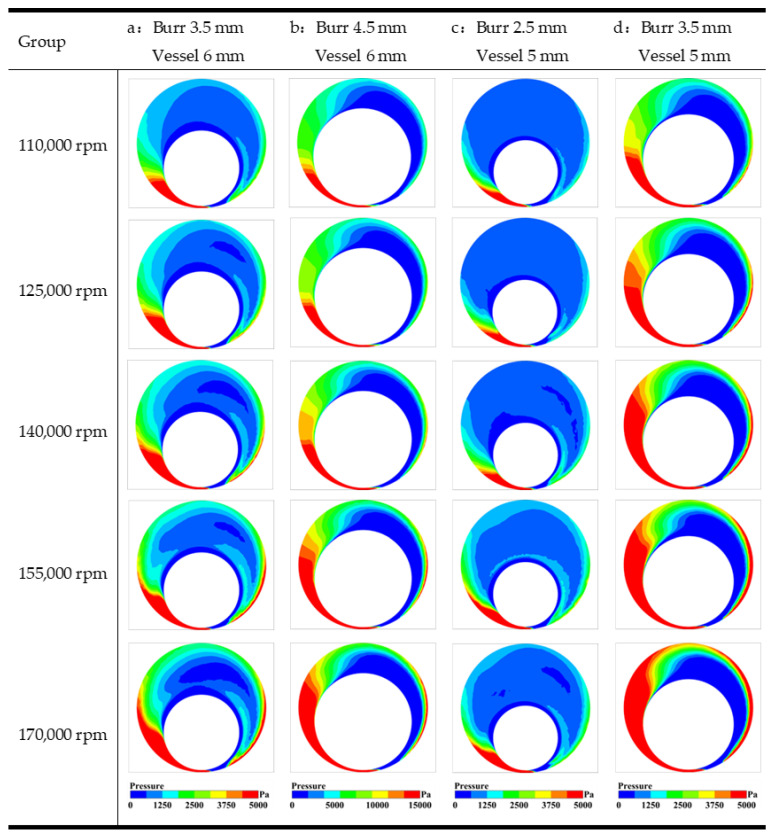
Pressure fields of different groups with different speeds. (**a**) burr 3.5 mm, vessel 6 mm; (**b**) burr 4.5 mm, vessel 6 mm; (**c**) burr 2.5 mm, vessel 5 mm; (**d**) burr 3.5 mm, vessel 5 mm.

**Figure 11 micromachines-14-02148-f011:**
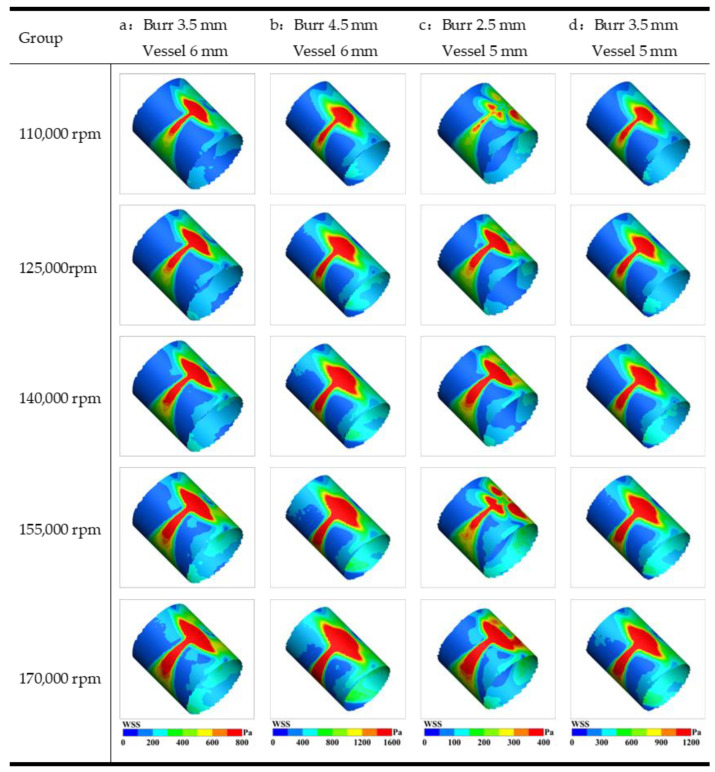
Stress forces of different groups with different speeds. (**a**) burr 3.5 mm, vessel 6 mm; (**b**) burr 4.5 mm, vessel 6 mm; (**c**) burr 2.5 mm, vessel 5 mm; (**d**) burr 3.5 mm, vessel 5 mm.

**Figure 12 micromachines-14-02148-f012:**
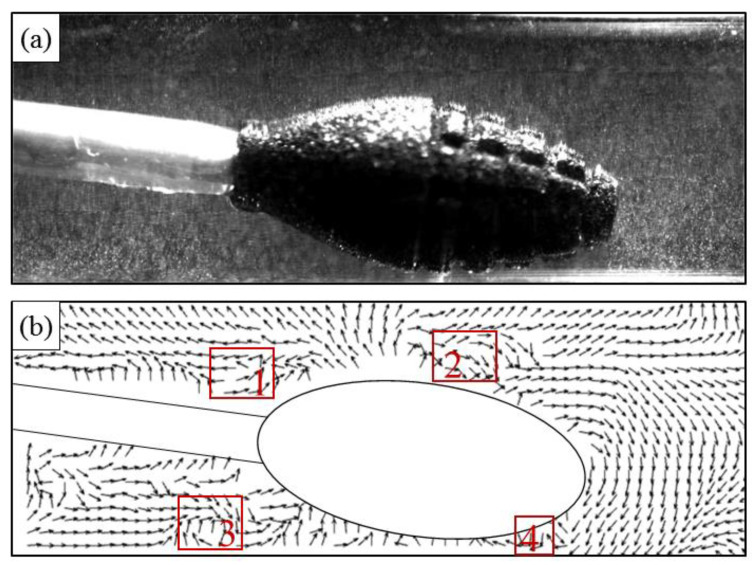
Experimental and simulation reference diagrams (position 1–4 are vortex areas). (**a**) experimental diagram; (**b**) simulation diagram.

**Table 1 micromachines-14-02148-t001:** Comparison table of experimental groups.

Group	Burr (B × L)/mm	Vessel (A)/mm	B/A
a	3.5 × 7	6	0.58
b	4.5 × 9	6	0.75
c	2.5 × 5	5	0.5
d	3.5 × 7	5	0.7

## Data Availability

Data are contained within the article.

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
