# Peer review of "Numerical Analysis of Stress Force on Vessel Walls in Atherosclerotic Plaque Removal through Coronary Rotational Atherectomy"

_micromachines, 2023, doi:10.3390/mi14122148_

Round 1
Reviewer 1 Report
Comments and Suggestions for Authors
This manuscript showcases the application of finite element simulation for analyzing wall shear stress patterns in medical procedures. Zhu et al offer valuable insights for optimizing these procedures and reducing potential complications. This article is excellent, and its data could have important implications for treating cardiovascular disease. I recommend submitting it for publication in MICROMACHINES. I have two minor suggestions: 1)improve the English writing; 2)simplify the content for better readability.
Comments on the Quality of English LanguageModerate editing of English language required
Author Response
Response to the Review Comments
Dear editor:
I would like to thank the editor and all reviewers for reviewing the paper (ID: micromachines-2650565, Title: Numerical analysis of stress force on vessel walls in atherosclerotic plaque removal by coronary rotational atherectomy)and providing valuable comments and opportunities to revise the paper for any deficiencies. These provided suggestions are crucial for improving the quality of this paper. We have carefully reviewed and addressed each suggestion made. All the changes made to the manuscript have been highlighted. Additionally, we have included a list of all comments and corresponding changes for the convenience of the reviewers. If the reviewers require further clarification on any aspect of the revised results, we would be glad to provide it. We hope to have further communication opportunities and thank the editors and reviewers for their time and effort in reviewing our manuscript.
Reviewer#1: This manuscript showcases the application of finite element simulation for analyzing wall shear stress patterns in medical procedures. Zhu et al offer valuable insights for optimizing these procedures and reducing potential complications. This article is excellent, and its data could have important implications for treating cardiovascular disease. I recommend submitting it for publication in MICROMACHINES. I have two minor suggestions: 1) improve the English writing; 2) simplify the content for better readability.
Response:
Thank you for your valuable and thoughtful comments. We value the quality of language expression and have made improvements to the text to ensure that our words can clearly and accurately convey what we want to express. We have appropriately simplified some of this manuscript to present our research more clearly and concisely.

Reviewer 2 Report
Comments and Suggestions for Authors
The study investigates the use of coronary rotational atherectomy for removing calcified tissue in cardiovascular disease using finite element simulation in an idealized model. However, the method appears to be incorrect. Please resubmit the study after conducting simulations with blood properties instead of water. Additionally, ensure that quantitative results are included in the abstract. When referring to "larger pressure," provide specific values, as "large" is too vague. Replace "finite element simulation" with "computational fluid dynamics" in the text.
Method:
Please provide justification for setting the speed at the inlet to 0.02 m/s and the pressure outlet to zero. Include details on the gap left between tools to enhance the reproducibility of the results. Given the expected rotational speed range of 110,000-170,000 rpm, consider increasing the interval for a more comprehensive study, at least with five different rpm settings. Three cases seem insufficient. Convergence criteria are usually set to 10^-6; clarify the rationale behind choosing 10^-4.
Results:
Consider moving the statement about "Wall shear stress is a crucial factor in arterial rupture" to the discussion section. Define "high speed" with specific values. The results section lacks quantitative data, so please enhance it. Specify how small the radial diameter is when referring to "smaller radial diameter." Verify whether the effect observed is shedding or vortices and consider adding a color bar to Figure 8.
Conclusion:
Include quantitative results in the conclusion section.
References:
Include more recent manuscripts from the past five years.
Minor:
Replace "he" with "the" in line 75. Leave a space between the number and unit when writing measurements.
Author Response
Response to the Review Comments
Dear editor:
I would like to thank the editor and all reviewers for reviewing the paper (ID: micromachines-2650565, Title: Numerical analysis of stress force on vessel walls in atherosclerotic plaque removal by coronary rotational atherectomy)and providing valuable comments and opportunities to revise the paper for any deficiencies. These provided suggestions are crucial for improving the quality of this paper. We have carefully reviewed and addressed each suggestion made. All the changes made to the manuscript have been highlighted. Additionally, we have included a list of all comments and corresponding changes for the convenience of the reviewers. If the reviewers require further clarification on any aspect of the revised results, we would be glad to provide it. We hope to have further communication opportunities and thank the editors and reviewers for their time and effort in reviewing our manuscript.
Reviewer#2:
1.The study investigates the use of coronary rotational atherectomy for removing calcified tissue in cardiovascular disease using finite element simulation in an idealized model. However, the method appears to be incorrect. Please resubmit the study after conducting simulations with blood properties instead of water. Additionally, ensure that quantitative results are included in the abstract. When referring to "larger pressure," provide specific values, as "large" is too vague. Replace "finite element simulation" with "computational fluid dynamics" in the text.
Response:
Thank you for your suggestions on the article. We have recalculated and submitted this study after simulations with blood properties instead of water. We have replaced some vague expressions with quantitative results in the abstract and replaced "finite element simulation" with "computational fluid dynamics" in the text.
Revised manuscript section:
The simulation results show that smaller ratios between burr and arterial diameter (B/A=0.5) result in a more stable flow field domain. Additionally, the pressure and stress force generated by the 4.5 mm diameter grinding tool reach 92.77 kPa and 10.36 kPa, surpassing those of the 2.5 mm and 3.5 mm grinding tools.
2.Method:
Please provide justification for setting the speed at the inlet to 0.02 m/s and the pressure outlet to zero. Include details on the gap left between tools to enhance the reproducibility of the results. Given the expected rotational speed range of 110,000-170,000 rpm, consider increasing the interval for a more comprehensive study, at least with five different rpm settings. Three cases seem insufficient. Convergence criteria are usually set to 10^-6; clarify the rationale behind choosing 10^-4.
Response:
Thank you for your suggestions on the article. We have further improved the parameter settings for simulation and provided corresponding basis. On the basis of 110000 rpm, 140000 rpm and 17000 rpm, 125000 rpm and 155000 rpm are added to achieve a more comprehensive study.
Revised manuscript section:
In these experiments, a 0.1mm clearance is left between the grinding tool and the cylindrical arterial wall to facilitate the removal of plaque from the inner wall of the vessel [18]. The blood flow velocity within the artery is highly rapid, averaging around 18-23 cm/s. The entrance speed is set at 20 cm/s to maintain a stable flow. The initial pressure at the outlet is set to zero to simulate a sufficiently long blood vessel. To study its effects comprehensively, the rotational speed is set to 110,000 rpm, 125,000 rpm 140,000 rpm, 155,000 rpm and 170,000 rpm.
For the residuals of the continuity, momentum, and turbulent transport equations, the convergence criterion is set at 10e-4 due to the exceptionally high speeds involved. Additionally, convergence is acknowledged if the flow difference between the entrance and exit is less than 1% of the entrance value.
3.Results:
Consider moving the statement about "Wall shear stress is a crucial factor in arterial rupture" to the discussion section. Define "high speed" with specific values. The results section lacks quantitative data, so please enhance it. Specify how small the radial diameter is when referring to "smaller radial diameter." Verify whether the effect observed is shedding or vortices and consider adding a color bar to Figure 8.
Response:
Thank you for your suggestions on the article. We have moved the statement about "Wall shear stress is a crucial factor in arterial rupture" to the subsequent discussion section. We have defined "high speed" or "smaller radial diameter" with specific values and added a color bar to Figure 8. Shedding is the periodic shedding of vortices caused by fluid on both sides behind the obstructing object. The vortex current generated by the obstruction of plaques at position 5 is Shedding.
Revised manuscript section:
The remaining vortices originate from the four primary vortices. It is noteworthy that the vortex region located downstream at position 5 represents shedding, a periodic shedding of vortices caused by fluid on both sides behind the obstructing object.
Figure 8. Vortex diagram of the same group with different speeds.
4.Conclusion:
Include quantitative results in the conclusion section.
Response:
Thank you for your suggestions on the article. We have modified the conclusion section with more quantitative results.
Revised manuscript section:
(3) The 4.5mm diameter grinding tool diameter produces the pressure of 92.77k pa during the rotational grinding process, which causes the container wall to form roughness.
(4) The shape and rotation direction of the grinding tool can influence the distribution and development of the stress force field. Compared to actual surgical procedures, the simulation results reveal that the stress force generated by fluid movement is 10.36k Pa, a value considerably lower than the actual measurement attributed to the absence of direct contact between the grinding tool and the simulated container wall in the fluid simulation.
5.References:
Include more recent manuscripts from the past five years.
Response:
Thank you for your suggestions on the article. We have replaced 6 references which were published within 5 years.
6.Minor:
Replace "he" with "the" in line 75. Leave a space between the number and unit when writing measurements.
Response:
Thank you for your suggestions on the article. We have replaced "he" with "the" in line 75 and invested more attention in the space between numbers and unit when writing measurements.

Reviewer 3 Report
Comments and Suggestions for Authors
The manuscript by Zhaoju Zhu et al. builted physical and simulation models of atherosclerotic plaque removal were conducted respectively to examine the velocity, vorticity, pressure, and stress fields of the boundary layer caused by the rotation of the grinding tool in a simulated container. The result provides valuable insights into the fluid dynamics of rotational grinding and its effects on vessel walls, which can serve as a reference for the development of safer and more effective surgical tools and techniques. However, this reviewer proposes that the manuscript could be considered the following points.
1. The effect of different sizes of atherosclerotic plaques on the outcome needs more discussion here.
2. Whether the material of the grinding tool itself will damage the vascular tissue?
3. Will the use of Will the use of abrasives cause vascular inflammation? IL-1β or IL-6 should be tested here. cause vascular inflammation? IL-1B or IL-6 should be tested here.
Author Response
Response to the Review Comments
Dear editor:
I would like to thank the editor and all reviewers for reviewing the paper (ID: micromachines-2650565, Title: Numerical analysis of stress force on vessel walls in atherosclerotic plaque removal by coronary rotational atherectomy)and providing valuable comments and opportunities to revise the paper for any deficiencies. These provided suggestions are crucial for improving the quality of this paper. We have carefully reviewed and addressed each suggestion made. All the changes made to the manuscript have been highlighted. Additionally, we have included a list of all comments and corresponding changes for the convenience of the reviewers. If the reviewers require further clarification on any aspect of the revised results, we would be glad to provide it. We hope to have further communication opportunities and thank the editors and reviewers for their time and effort in reviewing our manuscript.
Reviewer#3: The manuscript by Zhaoju Zhu et al. builted physical and simulation models of atherosclerotic plaque removal were conducted respectively to examine the velocity, vorticity, pressure, and stress fields of the boundary layer caused by the rotation of the grinding tool in a simulated container. The result provides valuable insights into the fluid dynamics of rotational grinding and its effects on vessel walls, which can serve as a reference for the development of safer and more effective surgical tools and techniques. However, this reviewer proposes that the manuscript could be considered the following points.
- The effect of different sizes of atherosclerotic plaques on the outcome needs more discussion here.
Response:
Thank you for your suggestions on the article. The primary focus of this manuscript is to explore the influence of varying rotational speeds and burr sizes on the flow field, pressure, and stress within the system. In this manuscript, plaques are represented as simplified raised ellipsoids. The size and morphology of plaques are also crucial and can be analyzed in subsequent studies. We hope to have further communication opportunities.
- Whether the material of the grinding tool itself will damage the vascular tissue?
Response:
Thank you for your suggestions on the article. There is differential cutting during the tool grinding process, where the tool only cuts diseased plaques without damaging normal vascular tissue. This is due to the different elastic moduli of plaques and blood vessels result in different cutting tools. I specifically have explained this phenomenon in the manuscript.
Revised manuscript section:
The hydrodynamic pressure film between the tool and the plaque is relatively thin, allowing direct contact and grinding of the plaque. Conversely, when the tool encounters the normal blood vessel wall, the dynamic pressure film between the tool and the vessel wall becomes thicker. This thicker film acts as a barrier, preventing the tool from grinding the normal blood vessel tissue.
- Will the use of abrasives cause vascular inflammation? IL-1β or IL-6 should be tested here.
Response:
Thank you for your suggestions on the article. This article primarily examines the hemodynamic characteristics of the tool's flow field, pressure, and stress during the rotary grinding process from an engineering standpoint. It employs numerical simulation calculations and experimental verification methods to conduct a comprehensive analysis. The use of abrasives may cause vascular inflammation such as vascular dissection, blood cell aggregation, and endometrial damage. The use of abrasives promotes tissue destruction and secretion of pro-inflammatory factors, including high levels of IL-1β and IL-6. Inflammation induced by IL-1β and IL-6 is a relatively specialized issue in the medical field, which can be further explored in subsequent research.

Round 2
Reviewer 3 Report
Comments and Suggestions for Authors
All of the previous reviews had been solved. I therefore recommend this paper to be publicized.